# Research on the enhancement path of green technology innovation efficiency under the group perspective

**Lei Liu, Li Zhang** **\*, Wei Xu**

School of Management, Shenyang University of Technology, Shenyang, Liaoning, China

\* zyali2023@163.com

## Abstract

China is at a critical moment of transforming high-speed development to high-quality development, and it is significant to improve the efficiency of green technological innovation. In this paper, under the perspective of two-stage innovation value chain, we construct the evaluation index system of green technology innovation efficiency, adopt the super efficiency SBM model to measure the green technology innovation efficiency of China's high-tech industries, and based on the results obtained, we assume the fuzzy set qualitative comparative analysis method (fs-QCA) based on the group theory to explore the complex causal mechanism and grouping paths of the interaction between enterprises, government and market that affects the green technology innovation efficiency Mechanism and group path. The study results show that (1) enterprise, government, and market are not necessary conditions to influence the efficiency of green technological innovation, and even if a particular party plays a central role, it needs the assistance of other parties. (2) The improvement of green technological innovation efficiency requires the interaction of enterprises, government, and market, and even if any party does not have the core conditions, it can still produce high green technological innovation efficiency. (3) The path of the "innovative compensation" effect is identified, which indicates that enterprises will generate a high level of green innovation efficiency under sufficient investment brought about by the enterprise scale effect and matched with a good level of economic development. (4) The market economy-led pathway suggests that when the market economy is highly developed, firms do not need environmental regulation and government support to generate efficient levels of green technological innovation.

## 1. Introductory

Forty years of reform, opening up, and massive investment have led to a huge breakthrough in China's economic development, growing from 0.3678 trillion yuan in 1978 to 100.8782 trillion yuan in 2020, with a growth rate of nearly 14.3%. The high-speed development of the economy has also brought problems such as rough economic structure, unbalanced development, weak technological innovation, serious waste of energy and resources, and pollution of the

**Data Availability Statement:** All relevant data are within the manuscript and its Supporting Information files.

**Funding:** Thank the Shenyang Philosophy and Social Sciences Special Fund (SYSK2023-JD-09)

for their support for this paper. Besides, the authors wish to acknowledge the contribution of the Liaoning Key Lab of Equipment Manufacturing Engineering Management, Liaoning Research Base of Equipment Manufacturing Development Liaoning Key Research Base of Humanities and Social Sciences. Research Center of micromanagement Theory, and Shenyang Association for Science and Technology.

**Competing interests:** The authors have declared that no competing interests exist.

environment. Green Transformation and Upgrading is the Way from High-Speed Development to High Quality and Sustained Development; technological innovation in the innovation process is constantly integrated into the green concept to bring technological progress and protect the ecological environment. In 2019, the National Development and Reform Commission and MSTC jointly published "Guidelines for Market-oriented Green Technology Innovation System." The text identifies the vital role of green technological innovation in economic development. It is essential to the 19th CPC National Congress, "Building a Green, Low Carbon, Recycling, and Sustainable Development." It is a necessary content of the 19th CPC National Congress "to build green, low-carbon, circular and sustainable development" [1]. Innovation in Green Technology is the Main Engine of Economic Transition. Therefore, the high-tech industry must strengthen support for green technology innovation and reduce resource consumption.

Green technological innovation is different from ordinary innovation. Innovation should generate knowledge and maximum economic benefits, reduce pollution emissions, save energy, protect the environment, and, more importantly, rely on innovation to achieve a harmonious symbiosis between society, the economy, and the environment [2]. To minimize the differences among regions, governments at all levels have introduced a series of policy measures; however, the results could be better. Therefore, policymakers should identify the many conditions that affect green innovation. They should also combine the region's helpful factors to propose a specialized, innovation-driven strategy for the area [3].

Scholars have explored the drivers of corporate green technology innovation in recent years from two perspectives, including external and internal drivers. Whether it is internal influencing factors or external influencing factors, the research has richer results. Researchers primarily conduct most current research from a micro point of view. Existing studies often take enterprises as the research object to study the impact of a single core variable on enterprises' green technological innovation or conduct regression analyses on several important factors affecting the efficiency of green technological innovation. They also explore the importance of the influencing factors by observing the significant level of each factor. As the factors affecting the efficiency of green technological innovation are the result of the interaction between multiple factors, there is a lack of research on the impact of antecedent conditions on the level of green technological innovation in linkage or group conditions, which in turn affects the judgment of the reasons for the differences in the efficiency of green technological innovation, limits the understanding of the synergistic effect of multiple factors such as systemic factors, market factors and the internal conditions of the enterprise itself, and further leads to ambiguity on the driving paths and the mechanism of the efficiency of green technological innovation [4]. Therefore, under a two-stage value chain perspective, this paper innovatively constructs a green technology innovation efficiency evaluation index system, which enriches it. It uses data envelopment analysis to measure efficiency and clarify the current level of green technology innovation in China's high-tech industries. Then, using fuzzy set qualitative comparative analysis, we screened multiple conditions from the three levels of market, government, and enterprise and analyzed how resources are mobilized among them to affect efficiency and how the multiple conditions are combined in different ways to enhance the efficiency of green technological innovation, it is conducive to mining the factors affecting the efficiency of green technological innovation and then correctly judging the reasons for the differences in green technological innovation. It reveals the role of different condition groupings on the level of green technological innovation. It provides solution ideas for clarifying the driving paths and role mechanisms of green technological innovation.

## 2. Literature review

### 2.1 Green technology innovation and its efficiency measurement

The idea was first put forward by Braun E D [5], who believed that an innovative process could be called "Green Innovation" if it could reduce environmental pollution or energy consumption. However, due to the different research perspectives, green innovation has been expressed differently in the academic world [6]. Many scholars believe that green innovation aims to make production efficient. It should minimize costs and maximize benefits. It does this, regardless of environmental conditions [7]. Some scholars consider the product life cycle and argue that green innovation is the whole process of continuously reducing the costs of the company and eventually moving to the product market, such as Yang Qingyi [8] will be the organization's innovation activities covered in the process of innovative product design, innovative production process, innovative organizational goals and innovative technology results of the green revolution is defined as "green innovation." As economic development consumes resources, scholars are paying more attention to 'green technology.' Most scholars believe that the core of green innovation is technological innovation, and green technological innovation has been more accurately described [9, 10]. Overall, green technological innovation is a deeper description of green innovation, which not only refers to the enterprise's innovation of products, processes, and production processes to improve efficiency and create more revenue but also requires that the innovation should protect the environment, and achieve the coordinated development of the three aspects of the society, the economy, and the environment [11].

The measurement of green technological innovation efficiency mainly consists of three approaches. One is to directly use the indicators that can represent green technology innovation, such as scholars Gao Xia, Zhihan, Zhang Fuyuan [1], Dong Zhiqing, and Wang Hui [12] use the green patent application volume or the green patent authorization volume, and other indicators that can represent the efficiency of green technology innovation. The second uses principal component analysis to measure corporate green technology innovation efficiency, such as Fan Qunlin [13], Wang Yurong [14] used Q-type cluster analysis to assess the level of green technology innovation in various regions of China and found that the eastern and central areas have a solid ability to introduce and absorb green technology; The third is to measure efficiency by establishing a green technology innovation input and output index system and using data envelopment analysis (DEA), which is widely recognized by scholars such as Lu Y H, Shen C C, Ting C T [15], who based on the traditional DEA approach, evaluates the efficiency of 194 high and new-technology enterprises in Taiwan; Lei L, LiLi Y [16] Measurement of Green Economic Efficiency in 30 Chinese Provinces by using Ultra Efficient SBM Model. However, few articles on the selection of indexes closely related to "green" exist when constructing evaluation indexes, so it is necessary to build a more reflective index system of green technology innovation efficiency to better reflect the status of green technological innovation.

### 2.2 Green technology innovation efficiency drivers

While researching the driving force behind green technological innovation, researchers have considered the factors of the company itself, such as human input [17], capital input [18], enterprise scale, internal redundant resources, corporate profitability, etc. Meanwhile, researchers also focus on the impact of external forces, such as market demand, stakeholders, and foreign direct investment at the market level [19], and The role of government-level factors such as environmental regulations, government support, government subsidies, and green credit in green technology innovation [20]. Zhou Zhong, Wang Ting, and Lu Haibo [21] measurd the innovation subject status of enterprises by the size of the enterprise, they found that

the continuous shrinking of the enterprise's size directly leads to the decline in the quality of the innovation subject. Dong Jingrong, Zhang Wenqing, and Chen Yuke [19] believe that in addition to the internal factors affecting the green innovation efficiency of enterprises, Many factors outside the enterprise affect green technology, such as some environmental regulations issued by the government to the enterprise to bring about the impact of the complexity of the enterprise. A reasonable choice of policy tools to ensure that the enterprise's technological innovation is incentivized should be a sensible choice Chen Dongjing and Leng Boyang [22] used the spatial Durbin model to test the role of three different environmental regulations on the efficiency of green technological innovation under the regulation of green credit. Other environmental regulations affect enterprises' green technological innovation and a reasonable combination of various policies is the key to improving the efficiency of enterprises' green technology innovation. In the study of the role of environmental regulation on the efficiency of green technological innovation. When exploring factors external to the enterprise, the role of the market has to be considered. The market is an invisible hand; the multi-dimensional market competition caused by opening up to the outside world effectively incentivizes enterprises to perform technical reform and product innovation. Yan Huafei Xiao Jing [23] constructed a middle model with a moderating effect. Then, they tested the effect of opening up to the outside world on the green innovation companies. Multiple dimensions, such as enterprise, government, and market, are essential.

By combining the concept of green technological innovation, the measurement method of efficiency, and the driving factors, it is found that the following shortcomings exist in the existing research. First, when scholars measure the efficiency of green technological innovation, they directly use the indicators instead of the level of green technological innovation, which is simple but not rigorous, or they use fewer fingers closely related to green when constructing the evaluation index system. The portrayal of green efficiency needs to be more detailed. Secondly, previous studies tend to study the impact of a single factor on the efficiency of green technological innovation from the perspective of a single factor, such as enterprises in the development of green technological innovation by the government's influence is more excellent, scholars on the impact of government regulation on the efficiency of green technological innovation is more in-depth research, research results are also more abundant. In contrast, the market and the impact of internal factors on the efficiency of green technological innovation of the enterprise's research results are relatively less, the lack of a systematic analysis of the efficiency of green technological innovation. There needs to be a more systematic understanding of green technology innovation efficiency. Thirdly, the existing research has also neglected the impact of the interaction or grouping of driving factors on the level of green technological innovation, especially the grouping effect generated by the linkage of the three levels of government-enterprise-market, which leads to ambiguity in the judgment of the mechanism of influencing factors on the efficiency of green technological innovation and the reasons for the differences in efficiency.

Therefore, this paper constructs an evaluation index system that can highlight "green" and measure the efficiency of green technical innovation using the super-efficient SBM model. With the help of the fuzzy stereotyped set comparative analysis method, we screen multiple conditions from the market, government, and enterprise levels, based on the group perspective, establish an analytical framework affecting the efficiency of green technological innovation, and analyze how to improve the efficiency of green technological innovation through different combinations under multiple conditions, and focus on how the driving factors of these three levels interact to form various groupings to affect the efficiency and reveal the roles of the groups of different conditions on the level of green technological innovation.

## 3. Theoretical model construction

From a group point of view, any result is a consequence of many factors. Therefore, the reasons why green technological innovation is highly inefficient or inefficient can also be attributed to many groupings. China is in the stage of transformation from high-speed development of high-quality development, and improving efficiency is the key. More than one factor is needed to promote the provinces' efficiency of green technological innovation. Thus, the impact of business, government, and the market on the level of green technological innovation is not a single one of these factors, but rather a synergistic one that works together.

### (1) Government level

From the government's point of view, to promote the efficiency of green technological innovation and give full play to the role of the government in the green technological innovation of enterprises, the government often pairs environmental regulatory policies with government support policies as a way to stimulate the company's willingness to innovate in a green way, regulate green innovative technologies, and thus enhance the company's efficiency in green innovation. However, when the government policy promotes the efficiency of green technology innovation, it can also solidify the model of green technology innovation or produce crowding-out effects. For example, for pollution-intensive enterprises, the promulgation of environmental regulations makes enterprises invest more costs in ecological management and end-product treatment, which leads to reduced funds invested in innovation and green innovation, resulting in the "crowding-out effect." In the face of "hard constraints," this type of enterprise often chooses to shut down or relocate. When the environmental regulation of corporate innovation compensation role is greater, the company's innovative behavior has sustainable development conditions, and the environmental regulation incentives for corporate innovation behavior, the enterprise can be sustainable development. Regarding the impact of government support and government regulation on the efficiency of green technological innovation, there are many research perspectives, and the research results are also rich; This literature has analyzed the subsidies for scientific research funding, technology subsidies, product price subsidies, and tax incentives, the results show that the relative strength of the "crowding-in effect" and "crowding-out effect" determines their direct effect on technological innovation and is affected by external factors such as government-enterprise relations, enterprise scale, enterprise R&D capability and willingness, and industrial characteristics.

### (2) Enterprise level

From the enterprise level, enterprise scale is undoubtedly a commonly recognized factor. Many scholars believe that expanding enterprise scale and improving enterprise profitability are critical factors in improving the efficiency of innovation. Expanding enterprise scale is not only an increase in personnel and resource elements but also an improvement in the internal organizational structure and the external environment of the enterprise. Some scholars believe that when the scale elements invested enterprises do not reach a certain amount, they will not produce or will not continue to produce innovation efficiency [24], so enterprises often expand scale elements, such as personnel and capital, to ensure the innovation output; some scholars believe that after the enterprise scale is enlarged to a certain degree, its impact on innovation efficiency will be weakened [25]; Some scholars believe that after the efficiency brought by the enterprise scale is maximized, the innovation ability of the enterprise will enter a state of depression or even decline [26]. However, the conclusions of domestic and foreign scholars differ on the most effective mechanism for enterprise size. Zhang Feng and other scholars [27] found that the size of the enterprise and its degree of innovation show an inverted

U-shaped pattern; That is, the size of the enterprise and its innovation efficiency are nonlinear functions. Voss G B, Sirdeshmukh D, Voss Z G [28], and other research are very representative of the field, that the size of the enterprise has a different impact on innovation and that the size of the enterprise has a different effect on innovation. Schwartz M, Peglow F, Fritsch M [29], and Liu Lei [30] found that enterprises' innovation efficiency is highly correlated with R&D expenditure and enterprise scale.

### (3) Market level

The essential feature of a market economy is market competition, and opening up to the outside world will intensify market competition. Opening up to the outside world mainly encompasses opening up a region to foreign trade and investment. With the opening of foreign exchange and foreign capital, under external pressure, local enterprises need to find or develop new technologies to maintain their market share; At this time, for sustainable development and cost considerations, most enterprises will choose to carry out technological innovation, product optimization, and innovation, to obtain competitive advantages, the resulting competitive effect will improve the efficiency of green technology innovation [31]. In addition, foreign trade enterprises have more advanced and foreign trade enterprises have more advanced and environmentally friendly technologies and substantial capital, which to a certain extent will produce a technology spillover effect, attracting local enterprises to imitate and learn to improve their green technology level. However, because of the existence of information asymmetry and other market failures, it is also easy to cause vicious competition, prompting enterprises to take unfair means to compete for market share and then reduce the enthusiasm of enterprises to carry out green technology R & D and innovation, which do not help increase the efficiency of technology innovation [32]. In addition to opening up to the outside world, environmental pollution also reduces the efficiency of green development; however, when the per capita income is increased, the degree of people's concern for the environment and their preference for green products will be increased, which will promote the enterprises to carry out green technological innovation to develop new markets. Technological innovation attracts investment from new markets and provides additional financial support to companies for green technological innovation.

To summarize, the efficiency of green technology innovation is influenced to varying degrees by factors at multiple levels rather than by a single or a few drivers alone; that is to say, for different regions, the weak level of a single element does not mean that the efficiency of their green innovation is low, other factors may highlight the story of their green technology innovation. For example, although some provinces are weak in independent R&D capability, enterprises are good at borrowing innovation models and technologies from other areas, which also realizes the improvement of innovation efficiency. The group effect of the linkages between the factors influencing the level of green technological innovation has become a theoretical fact that cannot be ignored, therefore, this paper tries to study the influence of the group effect on green technological innovation at the enterprise, government, and market levels, in which the enterprise level selects the two influencing factors of enterprise scale [27] and enterprise profitability, the market level is characterized by the degree of openness to the outside world [33] and the level of economic development. In contrast, the government level is characterized by the intensity of government environmental regulation [34] and government support [35]. The theoretical model is shown in **Fig 1**.

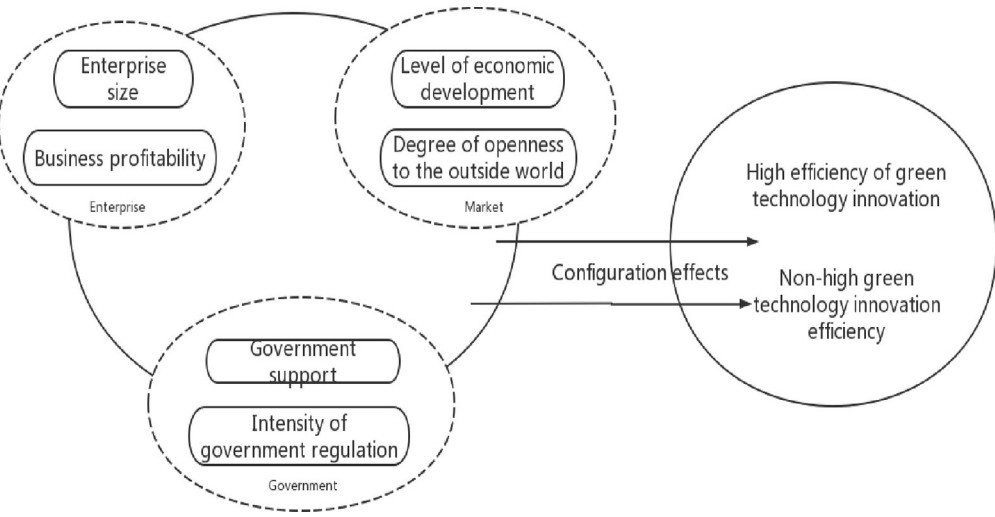

**Fig 1. Theoretical model of driving green technology innovation efficiency.**

## 4. Research methodology and data sources

### 4.1 Research methodology

First, this paper uses the super-efficient SBM model containing the non-expected outputs to measure China's high-technology industry's input and output efficiency. Second, based on the above measurements, the fuzzy qualitative set analysis method is adopted to explore the relationship between the driving factors and the level of green technological innovation. The reasons for choosing this method are as follows: ① Previous studies have primarily examined the effect of a single factor on efficiency [36], however improving the efficiency of green technology innovation depends on many factors; This method can reveal the synergistic effect, and interactive relationship between numerous factors and explore the impact of different paths on the level of green technological innovation. ② The marginal regression technique defaults to the symmetric relationship between independent and dependent variables; The impact of different driving factors of green technological innovation is not necessarily symmetric, and how to improve efficiency studied in this paper is an asymmetric relationship and, therefore, suitable for treatment with fs-QCA. ③ 30 provinces are selected as research samples in this study; This method is ideal for extensive sample analysis and suitable for small and medium-sized models. Thirty samples belong to medium pieces, and 4–7 conditional variables are selected for explanation, just in line with this method.

### 4.2 Data sources

The data from 30 provinces (excluding Tibet, Hong Kong, Macao, and Taiwan due to missing data) were selected as samples for analysis. The level of green technology innovation is the outcome variable, and the driving factors are the conditional variables. The data on the indicators of technology R&D and transformation of scientific and technological achievements involved in the calculation of the outcome variable mainly come from the China Statistical Yearbook on Science and Technology and the China Statistical Yearbook on Environment, among which the intermediate output indicators of the number of green invention patent applications and the number of green utility model patent applications come from the CnOpenData database. The indicators related to technology R&D, the transformation of scientific and technological

achievements, and the processing of the indicators are described in the second point of Chapter 5 of this paper (Selection of indicators).

Conditional variables are also from the China Statistical Yearbook on Science and Technology and the Technology and the China Statistical Yearbook on Environment. The indicators involved in the conditional variables and the processing of the indicator data are explained in detail in the conditional variance measurement in Chapter 5 of this paper. The missing data were supplemented with each city's statistical yearbooks and filled in using linear interpolation and mean-filling methods.

## 5. Variable measurement

### 5.1 Measurement of outcome variables

**5.1.1 Measurement model.** The DEA model is a typical representative of the nonparametric method, which is an evaluation method that calculates the relative efficiency by taking the optimal functional equation as a criterion. Tone [36] has studied the traditional DEA model. Finally, based on the DEA model, he proposed the super-efficient SBM model, which is a model that can further compare the efficiency value of the decision-making unit that has already reached efficiency. The formula of the super-efficiency SBM is as follows:

$$Z^* = \min \frac{\frac{1}{m} \sum_{i=1}^{m} \frac{a_i}{x_{i0}}}{\frac{1}{n+k} \left( \sum_{r=1}^{n} \frac{b_r}{y_{r0}} + \sum_{l=1}^{k} \frac{c_l}{p_{l0}} \right)}$$

$$s.t. \begin{cases} a \geq \sum_{j=1,\neq 0}^{J} \lambda_j x_j \\ b \leq \sum_{j=1,\neq 0}^{J} \lambda_j y_j \\ c \geq \sum_{j=1,\neq 0}^{J} \lambda_j b_j \\ a \geq x_0, b \leq y_0, c \geq p_0, b \geq 0, \lambda_j \geq 0 \end{cases}$$

where m, n, and k represent the number of input indicators, desired output, and non-desired output indicators, $x_{i0}$ denotes the input value, $y_{i0}$ denotes the desired output value, $p_{l0}$ denotes the non-desired output value, j denotes the province and city, a, b, and c all represent the slack variables,$\lambda$ is the weight vector, $z^*$ green technology innovation efficiency value, $z^*$ the larger, representing the higher level of green technology innovation efficiency.

**5.1.2 Selection of indicators.**

(1) *Input and output indicators for the R&D phase. Input indicators.* Enterprises commonly use R&D expenditure and R&D personnel to measure the level of their R&D investment. This paper uses personnel and funding to indicate inputs after drawing on the practice of Xiao Renqiao, Shen Lu, and Qian Li [37]. Therefore, this paper draws on the practice of Li Bin [38], Milani S [39], and other scholars, considering that regulations have a continuous impact, the number of local environmental protection laws, rules, and standards is accumulated year by year to represent command-and-control environmental regulation; With the

public becoming more and more concerned about environmental issues, the government and enterprises will take the lead in assuming the responsibility of environmental protection, which will lead to better public image and lower social costs; With the public becoming more and more concerned about environmental issues, the government and enterprises will take the lead in assuming ecological protection responsibilities, which will bring better public image and lower social governance costs. Therefore, this paper draws on the practice of Shim Shack J P, Ward M B [40] and uses the number of sudden environmental events to represent the public participation type of environmental regulation. In addition, Testa F, Iraldo F, and Frey M [41] found that market incentive-based environmental regulation requires enterprises to pay for the purchase of sewage rights, which is an additional cost burden for enterprises and will have a crowding out effect on the green development of enterprises. Hence, this paper adopts the amount of sewage charges levied to represent the market incentive-based environmental regulation.

*Output indicators*. The outputs obtained at the R&D stage are mainly based on scientific and technological knowledge, such as patents. Among all patents, invention patents and utility model patents are more representative of the output capacity of enterprises' scientific and technical achievements. Based on the previous research, this paper uses the WIPO Green List to screen green patents in the State Intellectual Property Office. It obtains the number of screened green invention and utility model patents from the database as the output indicators.

(2) *Input and output indicators for the transformation phase. Input indicators*. Technology transformation is how enterprises transform the scientific and technological achievements they have developed into tangible assets. The input indicators of this stage include innovative inputs and non-innovative inputs. Creative inputs are expressed by the outputs of the R&D stage, i.e., patents can verify whether the social and economic benefits they bring are good. In addition, based on green technology innovation to form green patents, this paper draws on the practice of Wu Zenghai and Li Tao [42]; it selects technological transformation, introduction fees, and year-end employees to represent the non-innovative inputs for technology transformation in this industry.

*Output indicators*. In this paper, the expected result in terms of economy - new product sales revenue - and the unanticipated result in terms of environmental benefits calculated using the entropy value method - composite environmental index - are selected, and both of them are used to measure the final result of green technological innovation. Among them, the new product sales revenue is the performance of the economy of the enterprise's green innovation results, which can reflect the economic contribution made to the enterprise by green technology. The comprehensive environmental index is a one-dimensional measurement index using the entropy value method to calculate industrial wastewater emissions, industrial sulfur dioxide emissions, and general waste solid disposal.

## 5.2 Conditional variable measurement

This study collects and organizes a large amount of related literature and, based on the research results of scholars, theoretically constructs the driving factors of green technology innovation efficiency and derives a theoretical model. Based on the group perspective, from the three levels of enterprise, government, and market, we explore the influence of six driving factors, namely, government support, environmental regulation intensity, economic development level, degree of opening up to the outside world, enterprise scale, and enterprise profitability, on the efficiency of green technological innovation, which is described as follows:

1. *Government support*: Scholars often use government funding and incentives as a cutoff point to examine their role in green technology innovation. Financial support includes financial inputs, such as fiscal inputs, tax levies, and technology subsidies, while incentive policies include tax incentives and innovation preferential policies. Government support can motivate enterprises to invest resource elements in green innovation projects and thus promote enhancing enterprises' green technology innovation efficiency [19]. For example, government R&D subsidies can enable enterprises to obtain liquidity in the short term, alleviate the dilemma caused by enterprise financing constraints, make enterprises feel optimistic about innovation, and increase their willingness to participate in creation. This paper adopts the research of Lv Yanwei, Xie Yanxiang, and Lou Xianjun [35]. It uses the proportion of government funds in the internal expenditure of industry R&D funding to indicate government support.

2. *Strength of environmental regulation*: Often initiated by governments, environmental regulation is a traditional means of managing environmental problems and is the main external driver of green technological innovation. However, due to the varying intensity of environmental regulation, its impact on the innovative behaviors of micro firms is still highly controversial. One view is that ecological regulations increase the costs of enterprises, crowd out their production resources, reduce their competitive advantages, and lead to the inhibition of technological innovation [43]; Another view is that the compensatory effect brought by appropriate environmental regulations can offset the increased costs of enterprises and get the benefits of innovation, also known as Porter's Hypothesis [44]. There are two ways to measure the intensity of environmental control; one is based on the perspective of governmental regulatory policies by selecting laws and regulations issued by the government, such as Smarzynska B W, Wei Shangjin [45], Fredriksson P G, Millimet D L [46], etc.; The other is that some scholars believe that after environmental regulations are implemented, the strength and scale of their enforcement are reflected by the amount of pollutants emitted, so the amount of each pollutant emitted is chosen as an indicator for measuring the intensity of environmental regulation, such as Cole M, Elliott R [47] and other scholars. In this paper, the power of ecological regulation is measured from the emission of various types of pollutants, and the emissions of the three wastes are selected. The comprehensive index method is applied to calculate them about the practice of Zhu Pingfang, Zhang Zhengyu, and Jiang Guolin [34]. The obtained indexes are summed up for each province's environmental regulation intensity. The formula follows where i denotes area and j denotes an emission item.

$$P_{ij} = \frac{X_{ij}}{\frac{1}{n}\sum_{i=1}^{m}X_{ij}}, (i = 1, 2, 3\ldots, mj = 1, 2, 3)$$

$$P_i = \frac{1}{3}(P_{i1} + P_{i2} + P_{i3})$$

3. *Level of economic development*: Generally speaking, the better a country's economic development, the greater its efforts to protect the environment. The level of economic growth and financial income of a region is often directly proportional to each other, and the higher the level of economic development of an area, the stronger its awareness of technological innovation, financial ability to pay, and environmental protection awareness compared to other regions [48]. Maybe because, with the improvement of innovation consciousness and

financial status, more resources will be invested to promote green technological innovation, introduce high-quality scientific research talents with more attractive treatment, give more priority to the introduction of advanced scientific research equipment, and gather more high-quality innovation factors to the field of green technological innovation, which will effectively promote the green technical innovation efficiency to be significantly improved, on the contrary, the weak level of economic development of the On the contrary, in places with an inadequate level of economic growth, the resource elements are not sufficient, resulting in low efficiency of green technology innovation. Therefore, the level of economic development is very relevant to the efficiency of green technological innovation, and according to previous studies, this paper chooses GDP per capita to express the level of economic development.

4. *Degree of openness to the outside world*: A higher degree of openness to the outside world can exacerbate market competition, enable enterprises to accelerate technological transformation, improve input-output relationships, and allocate resources more efficiently, thus indirectly improving the efficiency of green technological innovation. For example, with foreign direct investment as the carrier, many multinational corporations will transfer part of the industry to China so that China's local aggregation of high-level resources accelerates industrial transformation, improves employment, and enhances technological innovation efficiency. However, while opening up to the outside world, the low-cost manufacturing method attracts more foreign manufacturing industries for trade processing, and the increase of manufacturing export products predicts that it will bring higher resource consumption and environmental pollution, which in turn affects green technology innovation. In this paper, we refer to the practice of Yan Huafei and Xiao Jing [23] and use the ratio of exports of high-tech industries to GDP to indicate the degree of openness to the outside world.

5. *Enterprise scale*: The relationship between firm size and green technology innovation is in an inverted "U" shape, which means that, to a certain extent, The more significant the company size, the more positive the impact on innovation in green technologies. Therefore, appropriate enterprise-scale development helps improve the efficiency of green technology innovation [49]. However, as the company's size continued to expand, its efficiency did not improve further; it had a dampening effect, which is probably because the blind expansion of the enterprise led to increased management difficulties and neglected the improvement of efficiency [3]. We refer to the approach of Chi Renyong, Yu Jun, and Ruan Hongpeng [50], which describes enterprise size in terms of the ratio of primary business revenue and number of employees in high-tech industries.

6. *Enterprise profitability*: Enhancing enterprise capacity makes enterprises enter a virtuous cycle. That is, the profitability is improved, the enterprise has sufficient funds, the investment in research and development is increased, the enterprise carries out upgrading and transformation, the profitability is improved again, and in the process of continuous change and upgrading of the industry, the enterprise invests more in the green aspect. The proportion of the green sector has significantly increased, reducing environmental costs. This paper adopts the practice of Shaohua Wang, Zhiwei Yang, Ye Liu Wei Zhang Wei [51], which represents the enterprise's profitability by the sales profit, and the driving variable proxy variables are shown in **Table 1**.

**Table 1. Measurement of drivers of green technology innovation.**

| form | variant | Measurement method |
|---|---|---|
| governments | government support | Percentage of Government Funds in Internal Expenditures for Industry R&D Funding |
| | Intensity of environmental regulation | Composite index method for calculating three-waste emissions |
| corporations | Enterprise size | Revenue from main business/number of employees at the end of the period |
| | Corporate profitability | Profit on Sales = Net Profit / Sales Revenue |
| market | Level of economic development | GDP per capita |
| | degree of openness to the outside world | High-tech industry exports/GDP |

## 6 Calibration of variables and analysis of results

### 6.1 Calibration and description of variables

Based on Qu Xiaoyu and Qin Xutian's practice [52], we set 25, 50, and 75 percent of the sample describing the prior condition and the outcome as the anchor point for complete unaffiliated, cross-affiliated, and total affiliation, respectively, and the calibration is the process of assigning a pooled affiliation score to the cases in the range 0 to 1 [53]. The calibration and description results are presented using SPSS software to calculate the anchor. These are shown in **Table 2**.

### 6.2 Individual conditional necessity analysis

When analyzing the configuration of conditional variables, it is essential to first research the individual condition as necessary and verify whether that particular condition is needed for the outcome. This analysis is mainly measured by consistency and coverage, and it is now generally accepted in academia that when the surface of a condition is more significant than 0.9, it indicates that the condition must be present for the result to be obtained, i.e., when this outcome occurs, this condition is bound to happen. As seen from **Table 3**, The collinearity of each state variable is below 0.9, indicating that they are not necessary to achieve the efficiency of green technological innovation; therefore, it is necessary to explore the synergistic effects among the elements of green technological innovation.

### 6.3 Configuration path analysis

**6.3.1 Analysis of the grouping path of high green technology innovation efficiency.** To analyze the grouping relationship of different drivers, the calibrated data import into the

**Table 2. Calibration results.**

| variant | | | location | | | descriptive statistics | |
|---|---|---|---|---|---|---|---|
| | | | wholly unaffiliated | intersection point | wholly belong to | average value | standard deviation |
| outcome variable | efficiency | Green technology innovation efficiency | 0.2684 | 0.4798 | 0.7459 | 0.5734 | 0.3859 |
| conditional variable | market level | degree of openness to the outside world | 0.003 | 0.0222 | 0.0434 | 0.0290 | 0.0281 |
| | | Level of economic development | 10.8359 | 10.9767 | 11.2968 | 11.100 | 0.3685 |
| | Government level | government support | 0.0149 | 0.0268 | 0.0560 | 0.0034 | 0.0298 |
| | | Intensity of environmental regulation | 0.0033 | 0.0036 | 0.0038 | 0.0035 | 0.000 |
| | Enterprise level | Enterprise size | 149.8520 | 165.234 | 197.5985 | 173.55 | 47.578 |
| | | Corporate profitability | 0.0630 | 0.0811 | 0.1054 | 0.0923 | 0.0526 |

**Table 3. fs-QCA individual conditional necessity test.**

| Variant | High green technology innovation efficiency | | Non-high green technology innovation efficiency | |
|---|---|---|---|---|
| | Consistency | Degree of coverage | Consistency | Degree of coverage |
| Degree of openness to the outside world | 0.6950 | 0.6982 | 0.3860 | 0.3905 |
| ~ Degree of openness to the outside world | 0.3933 | 0.3889 | 0.7017 | 0.6984 |
| Level of economic development | 0.7010 | 0.6991 | 0.3920 | 0.3936 |
| ~ Level of economic development | 0.3920 | 0.3904 | 0.7003 | 0.7022 |
| Government support | 0.4515 | 0.4242 | 0.7050 | 0.6669 |
| ~Government support | 0.6455 | 0.6849 | 0.3913 | 0.4180 |
| Environmental regulatory intensity | 0.7109 | 0.5274 | 0.7544 | 0.5634 |
| ~Environmental regulatory intensity | 0.4115 | 0.6246 | 0.3672 | 0.5611 |
| Enterprise size | 0.5653 | 0.5768 | 0.4964 | 0.5099 |
| ~Enterprise size | 0.5197 | 0.5062 | 0.5880 | 0.5766 |
| Corporate profitability | 0.6308 | 0.6068 | 0.5196 | 0.5032 |
| ~Enterprise profitability | 0.4836 | 0.5000 | 0.5940 | 0.6183 |

software for grouping analysis; we set the threshold value of the sampling frequency to 1, the threshold for initial consistency to 0.8, and the PRI to 0.7, the intermediate solution, and the economical solution. We borrow from the practice of Ragin [54]; we determine the core conditions through the intermediate key and the economical solution. The grouping of driving elements of high green technological innovation efficiency is shown in **Table 4**. The table shows six groupings of high green technology innovation levels. The overall consistency of the solutions is 0.9581, which indicates that the six groupings are also sufficient conditions for realizing high green technology innovation in meeting the vast majority of cases. The overall coverage of the solutions is 0.49, which indicates that these groupings can explain 49% of the patients with high green technology innovation levels. In addition, a secondary equivalent structure can be constructed with consistent nuclear conditions. From Table 4, we can see that the core conditions of S2a and S2b are the same, so these two groupings can constitute a second-order grouping, S2, and the core conditions of S3a and S4a are the same, forming a second-order configuration of S3. As a result, this paper finally simplifies four grouping paths, as shown in **Table 4**. Typical examples of these pathways are shown in **Fig 2**.

① Multi-factor composite. Configuration S1 indicates that the market, the Government, and the enterprise tripartite without one party has the core conditions to Promote the benefits of green technology innovation and enterprise profitability, government support and other auxiliary conditions are also insufficient, at this time, the synergistic effect of the enterprise, the Government, and the market is a breakthrough to produce high innovation efficiency, with increased levels of openness, enterprise competition increases; To seize the market share, continuously improve the investment of R&D personnel and capital, and expand the scale of enterprises, it is necessary to consider its actual situation and reasonably use the Government's environmental control policy. On this basis, the multiple collaborations of enterprises, the Government, and markets provide a new impetus for green technological innovation and promote its efficiency with increased openness. In typical cases such as Gansu Province, to achieve sustainable development, Gansu Province's "14th Five-Year Plan" period customized a series of green development plans have been introduced to promote the development of green industry policies; With the Government's introduction of policies to guide changes in market demand, and then drive the green transformation of

**Table 4. Grouping of drivers of high green technology innovation efficiency.**

| variant | S1 | S2 | | S3 | | S4 |
|---|---|---|---|---|---|---|
| | | S2a | S2b | S3a | S3b | |
| degree of openness to the outside world | ● | ● | | ⊗ | ● | ● |
| Level of economic development | | ● | ● | ● | ● | ⊗ |
| government support | ⊗ | | ⊗ | ⊗ | ● | ⊗ |
| Intensity of environmental regulation | ● | ● | ● | ● | ⊗ | ● |
| Enterprise size | ● | ● | ● | ⊗ | ⊗ | ⊗ |
| Corporate profitability | ⊗ | ● | ● | ⊗ | ● | ● |
| consistency | 0.9564 | 0.9820 | 0.9669 | 0.9832 | 0.9091 | 0.9565 |
| original coverage | 0.1466 | 0.1827 | 0.2149 | 0.1177 | 0.1003 | 0.0882 |
| Unique coverage | 0.0221 | 0.0558 | 0.0872 | 0.0642 | 0.0569 | 0.0441 |
| Consistency of solutions | 0.9581 | | | | | |
| Coverage of solutions | 0.49 | | | | | |

● indicates that the condition exists and is important to the outcome, where ● indicates a core condition, ● indicates a marginal condition; a space indicates that the presence of the condition is not important to the outcome; ⊗Indicates that the condition is missing,⊗indicates that the core condition is missing,⊗indicates that the marginal condition is missing.

enterprises, We are increasing investment in science, technology, and innovation to promote the efficiency of green technology innovation.

② Enterprise size-market economy driven. Configuration S2a and S2b have the same core solution and constitute configuration S2. In contrast, the degree of openness, the intensity

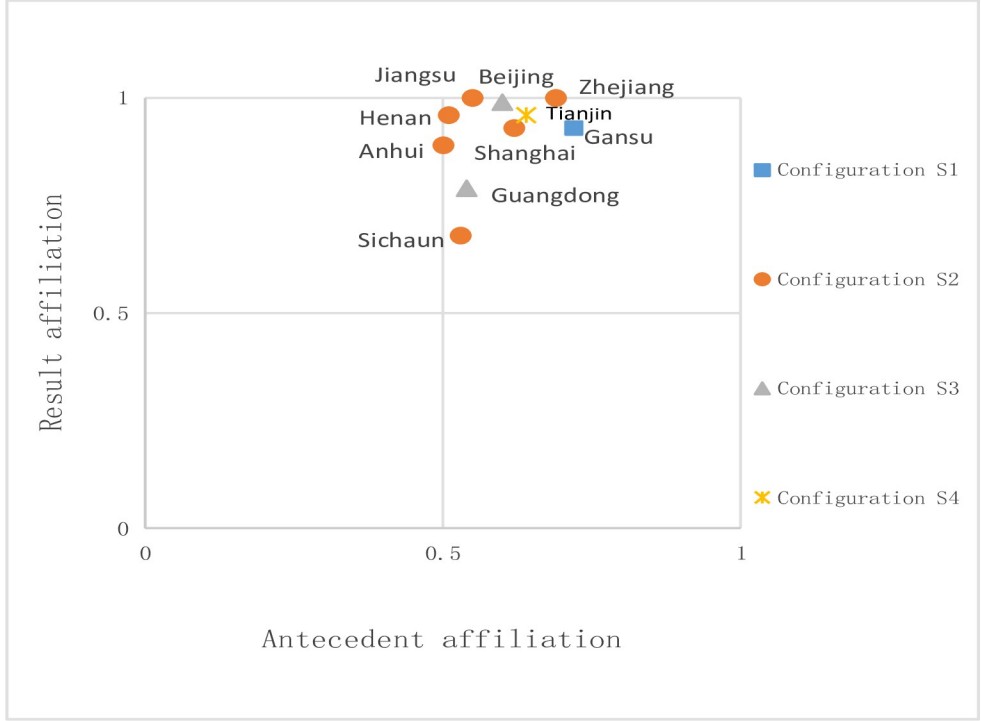

**Fig 2. Generate the configuration distribution of typical cases with high green technology innovation efficiency.**

of environmental regulation, and the profitability of enterprises are the marginal auxiliary factors that can improve the efficiency of green technology innovation. This grouping shows that in the case of large-scale enterprises and a high level of market economy development, even if the enterprise profitability is not enough or there is no government support, implementing environmental regulations and other policy systems can also improve the efficiency of green technology innovation. A mature market economy will shift resources to green enterprises. This will help green enterprises gain more market share; they will generate a larger enterprise scale and obtain more profits in a virtuous circle. Enough profits will let enterprises support green innovation. They can do this without government support or subsidies. They can do this even after paying the costs of environmental regulations. The path to green innovation is kept to a certain extent by the fact that This path supports Porter's Hypothesis to a certain extent because for the "innovative compensation" effect to occur, it needs to happen in the presence of large firms, i.e., firms with high R&D investment, and a high level of market economy development. This path is represented by Zhejiang, Anhui, Shanghai, Beijing, and Jiangsu, with Shanghai being the most typical. The Shanghai Municipal Commission of Economic and Informatization issued the "Fourteenth Five-Year Plan of Shanghai Municipal Industry Green Development" in 2022, which implies that Green development is an indispensable part of China's international competition. Shanghai, with a higher level of economic growth and openness to the outside world, shouldering the critical mission of leading the development of green innovation, from the creation of at least 200 green manufacturing demonstration units to promoting the demonstration area of the free trade zone green factory full coverage, from the expansion of the scale of the electric furnace steel furnace enterprises, to vigorously promote the upgrading of the petrochemical and chemical industries, and increase the investment of enterprises in green innovation, Shanghai fully utilize their advantages, the first to explore a new way in the country.

③ The market economy dominated. Similarly, the group states S3a and S3b have the same core solution and thus constitute a second-order group state S3, where the degree of economic development is essential in this case. Among them, the group state S3a indicates that in the point of the market not being open to the outside world, the market economy develops at a high level, the government does not adopt the support policy such as subsidy and strengthens the strength of environmental regulation, the enterprise can still produce high green technology innovation efficiency; S3b indicates that in the case of the market, the economy develops at a high level, the government removes the policy of environmental regulation and subsidizes the support to the enterprise, and with a certain degree of profitability and further opening up, the enterprise can still produce high green technology innovation efficiency. This path aligns with scholars who emphasize market-led efficiency improvements in The Transformation of China from Planned Economy to Socialist Market Economy; the innovation mechanism has also evolved from government-led, government-market-led to market-led. Introducing market orientation into the competitive process can effectively promote green technological innovation in this context. Therefore, the government should change from an innovation organizer to an innovation coordinator to create a good business environment and an orderly competitive environment. Guangdong and Beijing are typical representatives of this model. On 5 June 2020, the Beijing Municipal Development and Reform Commission issued the "Beijing Municipal Implementation Plan for Building a Market-Oriented Green Technology Innovation System," which puts forward the need to build a "market-driven" green technology innovation system. Markets have prompted more companies with green technology innovation capabilities to adopt traditional standards. At

the same time, Beijing has upgraded its new green technology innovation policy, supplemented with innovative green technology-related standards in several fields, and made sufficient preparations for improving green technology innovation.

④ Openness to the outside world driven. In group state S4, the degree of transparency to the outside world plays a significant role; At the same time, complementary environmental regulation intensity and enterprise profitability can lead to high efficiency, i.e., the degree of openness to the outside world is the core condition in this condition, which is the critical factor to generate high efficiency, and environmental regulation intensity and enterprise profitability are the marginal conditions, which are the auxiliary factors. This group states that the lack of economic development will result in lower efficiency of green technology innovation if we do not fully use the open policy; Therefore, when the development of the market economy is in the doldrums, enterprises should prioritize enhancing transparency and attracting foreign investment. This path supports the viewpoints of a series of scholars, such as Kong Lingchi [55], Wang Juan [32], Mac Dougall [56], Yan Huafei, and Xiao Jing [23]. Tianjin is a typical example of this path. In 2008, it created the Tianjin Central Eco-city; in 2013, it made the country's first "Green Development Demonstration Area"; And in 2019, it started the world's first "country-to-country" eco-city in the capacity of "model of international cooperation." In 2019, as a model of international cooperation, it created the world's first "eco-city" of synergistic development "between countries"; by 2020, foreign direct investment had reached 2.8 billion U.S. dollars. On this basis, Tianjin will continue to strengthen the regulation of new media and promote the development of green technology under the requirements of the Circular of the Secretariat Bureau of the General Office of the State Council on the Issuance of Inspection Indicators for Government Websites and New Media on Government Affairs, and Annual Assessment Indicators for Supervision Work.

**6.3.2 Grouping of non-high green technology innovation efficiency drivers.** In this paper, the group states that produce the non-high green technological innovation are also investigated to clarify the asymmetric causality resulting in green technology innovation heterogeneity. As can be seen from **Table 5**, the group state paths that generate non-high green technological innovation efficiency can be simplified into five. From group state S1, it can be seen that in the case of a weak level of economic development and a small enterprise scale, the enterprise's profitability is poor, which will lead to the emergence of non-high green technological innovation efficiency. S2 further explains that the weak level of economic development, even if the enterprise has the support of the government and continues to expand the enterprise scale, the enterprise itself has poor profitability situation, the green innovation efficiency is still not improved.

ConfigS3 further argues for the critical role of markets in enhancing the efficiency of green technology innovation. When the market is not developed, i.e., the market is not open to the outside world, and the market economy is not active, the efficiency of green technological innovation cannot be improved even if the enterprise has strong profitability. Suppose the economy is well developed, and the government strongly supports environmental regulation to ensure ecological protection efficiency. Can it improve the level of green technological innovation? Configuration S4 answers. Configuration S4, especially S4b, shows that high economic development level, government support and environmental regulation, and high profitability of enterprises still cannot improve the efficiency, in which case the advantages of enterprises, the government, and the market all have become the core conditions that reduce the efficiency. The excessive participation of all parties leads to scattered, duplicated, and complicated resources. To get more government support, enterprises apply for various subsidies. Still, the

**Table 5. Grouping of drivers of innovation efficiency in non-high green technologies.**

| variant | S1 | S2 | S3 | | S4 | | S5 |
|---|---|---|---|---|---|---|---|
| | | | S₃ₐ | S₃ᵦ | S₄ₐ | S₄ᵦ | |
| degree of openness to the outside world | ⊗ | ⊗ | ⊗ | ⊗ | ⊗ | • | ⊗ |
| Level of economic development | ⊗ | ⊗ | ⊗ | ⊗ | ● | ● | ⊗ |
| government support | | ● | ⊗ | ⊗ | ● | ● | • |
| Intensity of environmental regulation | ⊗ | • | • | ⊗ | ● | ● | • |
| Enterprise size | ⊗ | ● | ⊗ | • | ⊗ | • | |
| Corporate profitability | ⊗ | | ● | ● | ● | ● | ⊗ |
| consistency | 0.8599 | 0.9607 | 0.8921 | 0.8644 | 0.8313 | 0.8040 | 0.9763 |
| original coverage | 0.2079 | 0.2758 | 0.0823 | 0.0677 | 0.0916 | 0.139 | 0.2731 |
| Unique coverage | 0.0684 | 0.0372 | 0.0412 | 0.0033 | 0.0399 | 0.0871 | 0.022 |
| Consistency of solutions | 0.876 | | | | | | |
| Coverage of solutions | 0.5915 | | | | | | |

government adheres to the principle of "earmarking," which makes the expenditure and demand unequal and often creates a dilemma where there is no money to use, money is obtained, and no land. In addition, in the process of innovation, enterprises, the government, and the market on the positioning of innovation efficiency assessment is also very different; different subjects on the evaluation of innovation efficiency often have their considerations, and finally, innovation efficiency is low. Grouping S5 shows that only government involvement and non-core conditions still produce low green technology innovation efficiency. The typical case is shown in **Fig 3**.

## 6.4 Robustness check

In this paper, two methods are used to test the robustness of the results. The first method is to use the site-specific test to adjust the threshold for determining the entire affiliation point from '0.75' to '0.95', and the histogram results are unchanged; Then, referring to the practice of Zhang Ming and Du Yunzhou [57], the original consistency threshold is adjusted from 0.8 to 0.85, and the histogram results are mostly consistent, and individual histograms are changed, as shown in **Table 6**. The main path of high green technology innovation efficiency is composed of the degree of openness to the outside world, the level of economic development, and the size of the enterprise as the core while assisting other marginal conditions, which is not much different from the results above; The robustness test also identifies paths such as S3, S1, in the non-high grouping state. The second method uses statistically specific methods to conduct stability tests across different periods. In this paper, data from the year under the condition variable and the outcome variable were selected for the histogram analysis, aiming to compare the results produced by different years of data for the same indicator, as shown in **Table 7**, the results of the histograms are primarily consistent, identifying three paths, S2, S3, and S4, that are high in the efficiency of green technological innovations, and also identifying the S1, S3, and S4 paths in the non-high group state—the results of the robustness test show that this study has good robustness.

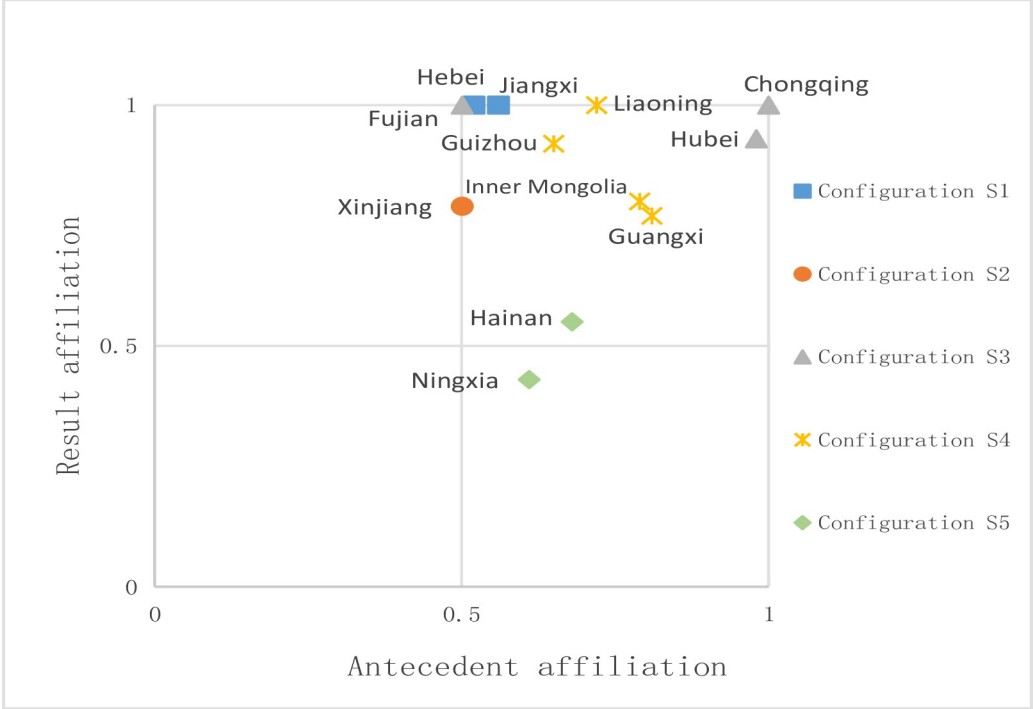

**Fig 3. Typical case configuration distribution of non-high green technology innovation efficiency.**

## 7 Conclusions and implications

### 1. Conclusion

First, an indicator system is established to measure the efficiency of green technology innovation in enterprises. Based on the above, the paper discusses the mechanism and how to

**Table 6. Robustness test results.**

| variant | High group robustness results | | | | | | Non-high-group robustness results | | | |
|---|---|---|---|---|---|---|---|---|---|---|
| | S1 | | S2 | S3 | S4 | | S1 | S2 | S3 | |
| | $S_{1a}$ | $S_{1b}$ | | | $S_{4a}$ | $S_{4b}$ | | | $S_{3a}$ | $S_{3b}$ |
| degree of openness to the outside world | ● | ● | | ⊗ | ● | ● | ⊗ | ⊗ | ⊗ | ⊗ |
| Level of economic development | | • | ● | ● | ⊗ | • | ⊗ | ⊗ | ⊗ | ⊗ |
| government support | ⊗ | | ⊗ | ⊗ | ⊗ | • | ● | ● | ⊗ | ⊗ |
| Intensity of environmental regulation | • | • | • | • | • | ⊗ | | ● | • | ⊗ |
| Enterprise size | ● | ● | ● | • | ⊗ | ⊗ | • | • | ⊗ | • |
| Corporate profitability | ⊗ | ⊗ | • | • | • | • | ⊗ | | ● | ● |
| consistency | 0.9564 | 0.9820 | 0.9669 | 0.9832 | 0.9565 | 0.9091 | 0.9664 | 0.9607 | 0.8921 | 0.8644 |
| original coverage | 0.1466 | 0.1827 | 0.2149 | 0.1177 | 0.0883 | 0.1003 | 0.1913 | 0.2757 | 0.0824 | 0.0677 |
| Unique coverage | 0.0221 | 0.0576 | 0.0871 | 0.0642 | 0.0441 | 0.0569 | 0.0425 | 0.0918 | 0.0412 | 0.0033 |
| Consistency of solutions | 0.9581 | | | | | | 0.9312 | | | |
| Coverage of solutions | 0.4899 | | | | | | 0.3688 | | | |

**Table 7. Robustness test results.**

| variant | High group robustness results | | | | | Non-high-group robustness results | | | | |
|---|---|---|---|---|---|---|---|---|---|---|
| | S1 | S2 | | S3 | S4 | S1 | S2 | | S3 | S4 |
| | | S2a | S2b | | | | S2a | S2b | | |
| degree of openness to the outside world | ● | ⊗ | | ⊗ | ● | ⊗ | ● | ● | ⊗ | ⊗ |
| Level of economic development | ● | | • | ● | ⊗ | ⊗ | | • | ● | ⊗ |
| government support | ⊗ | ● | ● | ⊗ | | | ● | ● | ● | ⊗ |
| Intensity of environmental regulation | • | ⊗ | ⊗ | • | • | • | • | • | ⊗ | ⊗ |
| Enterprise size | ● | • | • | ⊗ | • | • | ⊗ | ⊗ | ⊗ | • |
| Corporate profitability | | ⊗ | ⊗ | • | ⊗ | ⊗ | ⊗ | | ● | ● |
| consistency | 0.8701 | 0.9888 | 0.9876 | 0.9054 | 0.9448 | 0.9125 | 0.9651 | 0.8691 | 0.9578 | 0.8731 |
| original coverage | 0.1615 | 0.1158 | 0.1041 | 0.1308 | 0.0381 | 0.1782 | 0.1325 | 0.1270 | 0.1086 | 0.0799 |
| Unique coverage | 0.1393 | 0.0410 | 0.0351 | 0.1054 | 0.0410 | 0.1243 | 0.0286 | 0.0382 | 0.0560 | 0.0539 |
| Consistency of solutions | 0.9213 | | | | | 0.8958 | | | | |
| Coverage of solutions | 0.4498 | | | | | 0.4228 | | | | |

improve the efficiency of green technology innovation using the Fuzzy Set method. The following conclusions are reached:

1. None of the six elements at the three levels of enterprise, government, and market can serve as the sole condition for enhancing the efficiency of green technological innovation. It is a complex process to improve the efficiency of green technological innovation under the joint action of multiple forces such as enterprises, governments, and markets.

2. There are four significant ways to generate high levels of green technology innovation, one of which is to find ways to improve the efficiency of green technological innovation in the absence of core conditions through tripartite cooperation between business, government, and the market, which reveals that cooperation among the three is essential. The second is centered on the market economy and enterprise scale, supplemented by marginal conditions such as the intensity of environmental control, to achieve higher efficiency in the innovation of green technologies, i.e., where the level of development of the market economy and the size of the undertaking is significant, an increase in the level of environmental protection may be envisaged, and in conjunction with the actual situation of the enterprise, to select an appropriate path to enhance the enterprise's green technological innovation capacity. The third one is market economy-led; when the scale effect of enterprises is not well exerted, the path with the market economy as the core can be considered, and government support or one of the government's environmental regulations can be selected as an assistant condition to increase the efficiency of green technology innovation. The last one is open-driven, with openness as the core, complementing the intensity of environmental regulations and corporate profitability to improve efficiency.

3. There are several scenarios in which non-high green technology innovations can arise: ① Lack of enterprise size, profitability, and market economy development;② When the level of market economic development is missing, and only government support and enterprise scale are core conditions; ③ When the size of the firm and the degree of openness to the outside world are not central conditions, and all other variables are central conditions; ④ when all variables are core conditions, except for the size of the enterprise and openness to

the outside world; ⑤ when the core conditions are missing, and government support and the strength of environmental regulation are marginal conditions.

## 2. Insights

1. The improvement in the innovation efficiency of green technology is due to the multi-body linkage; no single party can dominate, especially without the core conditions. It is necessary to consider the interactive effect of multi-body and realize the high efficiency of technology innovation. When the enterprise's green technology innovation efficiency is higher, we must not unquestioningly optimize to strengthen the impact of a single element; we need to focus on all matching aspects and make combinations that can increase efficiency. Each region lacks resources, but following their situation and all the advantages, combined with the market, policy, and other factors, to explore and develop a path of green development in line with the local.

2. The level of regional economic development is fundamental to achieving green science, technology, and innovation. The higher level of economic growth in the region has more funds to support innovation, supported enterprises to expand the scale of enterprises, and then attract a variety of social capital to participate in green creation, to market funds as part of the source of funds for enterprise innovation. Therefore, enterprises should fully utilize the invisible hand of the market, together with the use of government policies, to promote the formation of market funds to support the construction of the green innovation chain of enterprises so that the market is deeply involved in the R&D and transformation of green technological innovation in all aspects.

3. Increasing the opening-up level will help improve the efficiency. Introducing advanced innovative infrastructure can create a collaborative innovation platform, offering the best hardware support to improve technology innovation. More foreign capital can be brought in to bring in sufficient and advanced innovation resources and concepts, which can enhance local innovation. At the same time, the more open the region is, the more innovative capital and talented people will be attracted to it. Therefore, given the insufficient level of economic development, enhancing the degree of openness to the outside world, in combination with the existing environmental control policies, and utilizing the dividends of openness to bring in advanced innovative equipment and innovative talents is an effective way to enhance the profitability of enterprises and improve the innovation capacity of green technology.

## 3. Limitation

The limitations of this paper are mainly reflected in the following: Firstly, limited to the number of variables that can be handled by fs-QCA, only six factors affecting the efficiency of technological innovation have been selected, and other factors have not been included thus. They fail to present a complete causality and explore more informed conclusions and laws. In future research, we can consider using questionnaires, surveys, and other methods to design more comprehensive evaluation indicators. Secondly, this paper only focuses on target companies from high-tech industries, and the findings may apply poorly to other industries. Future researchers can collect data from other industries to compare to the findings of this paper. Future research can further explore the above shortcomings.

## Supporting information

**S1 Data. Raw data.**
(XLS)

**S1 File. Published papers.**
(PDF)

## Acknowledgments

**Notes on contributors**

Lei Liu, Master, Associate Professor, School of Management, Shenyang University of Technology. His research interests include science and technology innovation and management.

Li Zhang, Master candidate, School of Management, Shenyang University of Technology. Her research interests include science and technology innovation and management.

Wei Xu, Doctor, Professor, School of Management, Shenyang University of Technology. His research interests include operations management and industrial engineering.

## Author Contributions

**Conceptualization:** Lei Liu, Li Zhang.

**Data curation:** Li Zhang.

**Formal analysis:** Li Zhang.

**Funding acquisition:** Lei Liu, Wei Xu.

**Investigation:** Li Zhang.

**Methodology:** Lei Liu, Li Zhang.

**Project administration:** Li Zhang, Wei Xu.

**Resources:** Lei Liu, Li Zhang.

**Software:** Li Zhang, Wei Xu.

**Supervision:** Lei Liu, Li Zhang, Wei Xu.

**Validation:** Li Zhang.

**Visualization:** Li Zhang.

**Writing – original draft:** Li Zhang.

**Writing – review & editing:** Lei Liu, Li Zhang.

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
