## [Decision Letter · Decision Letter 0]

11 Dec 2023

PONE-D-23-35015Research on the Enhancement Path of Green Technology Innovation Efficiency under the Group PerspectivePLOS ONE

Dear Dr. Zhang,

Thank you for submitting your manuscript to PLOS ONE. After careful consideration, we feel that it has merit but does not fully meet PLOS ONE’s publication criteria as it currently stands. Therefore, we invite you to submit a revised version of the manuscript that addresses the points raised during the review process.

We look forward to receiving your revised manuscript.

Kind regards,

Yang Gao

Academic Editor

PLOS ONE

Journal Requirements:

2. Thank you for submitting the above manuscript to PLOS ONE. During our internal evaluation of the manuscript, we found significant text overlap between your submission and previous work in the [introduction, conclusion, etc.].

Please revise the manuscript to rephrase the duplicated text, cite your sources, and provide details as to how the current manuscript advances on previous work. Please note that further consideration is dependent on the submission of a manuscript that addresses these concerns about the overlap in text with published work.

[If the overlap is with the authors’ own works: Moreover, upon submission, authors must confirm that the manuscript, or any related manuscript, is not currently under consideration or accepted elsewhere. If related work has been submitted to PLOS ONE or elsewhere, authors must include a copy with the submitted article. Reviewers will be asked to comment on the overlap between related submissions (http://journals.plos.org/plosone/s/submission-guidelines#loc-related-manuscripts).]

We will carefully review your manuscript upon resubmission and further consideration of the manuscript is dependent on the text overlap being addressed in full. Please ensure that your revision is thorough as failure to address the concerns to our satisfaction may result in your submission not being considered further.

"2022 Shenyang Philosophy and Social Science Planning Project, "Research on the Path and Countermeasures of Digital Economy to Drive Shenyang's Industrial Structure Upgrading" (SY202237Y)"

"Thank the 2022 Shenyang Philosophy and Social Science Planning Project, "Research on the Path and Countermeasures of Digital Economy to Drive Shenyang's Industrial Structure Upgrading" (SY202237Y), for supporting this project."

"2022 Shenyang Philosophy and Social Science Planning Project, "Research on the Path and Countermeasures of Digital Economy to Drive Shenyang's Industrial Structure Upgrading" (SY202237Y)"

6. Thank you for stating the following in your Competing Interests section:  

"The authors declare that they have no known competing financial interests or personal relationships that could have appeared to influence the work reported in this paper."

7. In your Data Availability statement, you have not specified where the minimal data set underlying the results described in your manuscript can be found. PLOS defines a study's minimal data set as the underlying data used to reach the conclusions drawn in the manuscript and any additional data required to replicate the reported study findings in their entirety. All PLOS journals require that the minimal data set be made fully available. For more information about our data policy, please see http://journals.plos.org/plosone/s/data-availability.

8. PLOS requires an ORCID iD for the corresponding author in Editorial Manager on papers submitted after December 6th, 2016. Please ensure that you have an ORCID iD and that it is validated in Editorial Manager. To do this, go to ‘Update my Information’ (in the upper left-hand corner of the main menu), and click on the Fetch/Validate link next to the ORCID field. This will take you to the ORCID site and allow you to create a new iD or authenticate a pre-existing iD in Editorial Manager. Please see the following video for instructions on linking an ORCID iD to your Editorial Manager account: https://www.youtube.com/watch?v=_xcclfuvtxQ

Reviewers' comments:

Reviewer's Responses to Questions

**Comments to the Author**

1. Is the manuscript technically sound, and do the data support the conclusions?

Reviewer #1: Yes

2. Has the statistical analysis been performed appropriately and rigorously? 

Reviewer #1: Yes

3. Have the authors made all data underlying the findings in their manuscript fully available?

Reviewer #1: Yes

4. Is the manuscript presented in an intelligible fashion and written in standard English?

Reviewer #1: Yes

5. Review Comments to the Author

Reviewer #1: Overview of the manuscript

The work focuses on the improving green technology innovation efficiency. In the work the authors use the super efficiency SBM model to measure the green technology innovation efficiency of high-tech industries, assuming the fuzzy set qualitative comparative analysis method (fs-QCA) based on the grouping theory to explore the complex causal mechanism of the interaction between enterprises, government, and the market. The study shown that three factors enterprise, government, and market are not a single necessary condition to influence the efficiency of green technological innovation, and the improvement of green technological innovation efficiency requires the interaction of enterprises, government, and market. Furthermore, the authors identify the path of the "innovation compensation" effect which indicates that enterprises will generate a high level of green innovation efficiency under sufficient investment that matched with a good level of economic development, and when the market economy is highly developed, firms do not need to rely on environmental regulation and government support but rather on one or the other to generate high green technology innovation efficiency.

GENERAL COMMENT

The work is interesting, the topic discussed is highly innovative and well matches social, economic and environmental interests. The topic is extensively developed in terms of its problem and a broad overview on the problem solving related to the topic is offered. The mathematical-analytical procedure is rigorous and offer readers a methodological procedure that can also be applied in other similar contexts. The bibliographic analysis is accurate and relevant to work purpose.

The work is specifically oriented towards experience and reality related to Chines perspective; this may encounter occasional difficulties in replicating the conclusion of the work in different realities.

This specific application should be better highlighted in the Introduction section.

SPECIFIC COMMENTS

Pag. 1, line 11-19: the paragraph is too long and syntactically complex. Make shorter sentences.

Pag. 2, line 38-39: the sentence is not clear “three aspects of society, economy, and environment.” And the third? Rephrase the sentence or explain better.

6. PLOS authors have the option to publish the peer review history of their article (what does this mean?). If published, this will include your full peer review and any attached files.

Reviewer #1: No

---

## [Author Response · Author response to Decision Letter 0]

17 Jan 2024

Dear reviewers

Thank you for reviewing our manuscript, we greatly appreciate your constructive suggestions and have revised the manuscript accordingly. We have now clarified the applicability of the manuscript in more detail and have carefully revised the relevant wording issues you raised.The specific changes can be found in the "Response to Reviewer" document.

---

## [Decision Letter · Decision Letter 1]

13 Mar 2024

PONE-D-23-35015R1Research on the Enhancement Path of Green Technology Innovation Efficiency under the Group PerspectivePLOS ONE

Dear Dr. Zhang,

Thank you for submitting your manuscript to PLOS ONE. After careful consideration, we feel that it has merit but does not fully meet PLOS ONE’s publication criteria as it currently stands. Therefore, we invite you to submit a revised version of the manuscript that addresses the points raised during the review process.

We look forward to receiving your revised manuscript.

Kind regards,

Tinggui Chen

Academic Editor

PLOS ONE

Journal Requirements:

Additional Editor Comments:

I have completed my evaluation of your manuscript. The reviewers recommend reconsideration of your manuscript following minor revision. I invite you to resubmit your manuscript after addressing the comments below.

Reviewers' comments:

Reviewer's Responses to Questions

**Comments to the Author**

1. If the authors have adequately addressed your comments raised in a previous round of review and you feel that this manuscript is now acceptable for publication, you may indicate that here to bypass the “Comments to the Author” section, enter your conflict of interest statement in the “Confidential to Editor” section, and submit your "Accept" recommendation.

Reviewer #1: All comments have been addressed

Reviewer #2: All comments have been addressed

2. Is the manuscript technically sound, and do the data support the conclusions?

Reviewer #1: Yes

Reviewer #2: Yes

3. Has the statistical analysis been performed appropriately and rigorously? 

Reviewer #1: Yes

Reviewer #2: Yes

4. Have the authors made all data underlying the findings in their manuscript fully available?

Reviewer #1: Yes

Reviewer #2: Yes

5. Is the manuscript presented in an intelligible fashion and written in standard English?

Reviewer #1: Yes

Reviewer #2: Yes

6. Review Comments to the Author

Reviewer #1: The authors revised their work, accepting my suggestions.

The manuscript has significantly improved in its presentation.

No more concerns

Reviewer #2: 1. The authors are expected to add references to support the literature review section and not write from their own knowledge and experience.

2. The authors should have expressed more clearly the theoretical and practical implications of this study.

3. The authors should have added more compelling information in the data sources section.

4. The writer should revise the language of the paper as a whole; there are many grammatical errors in this paper. It is recommended that the writer seek the help of a native English speaker.

7. PLOS authors have the option to publish the peer review history of their article (what does this mean?). If published, this will include your full peer review and any attached files.

Reviewer #1: No

Reviewer #2: No

---

## [Author Response · Author response to Decision Letter 1]

12 Apr 2024

Thank you for reviewing our manuscript, we greatly appreciate your constructive suggestions and have revised the manuscript accordingly. We have now clarified the applicability of the manuscript in more detail and have carefully revised the relevant wording issues you raised. And, we have carefully examined the manuscript and carefully revised similar issues that have arisen elsewhere. Thank you again for your valuable suggestions. You can see the exact changes in the revised version.

---

## [Decision Letter · Decision Letter 2]

30 Apr 2024

PONE-D-23-35015R2Research on the Enhancement Path of Green Technology Innovation Efficiency under the Group PerspectivePLOS ONE

Dear Dr. Zhang,

Thank you for submitting your manuscript to PLOS ONE. After careful consideration, we feel that it has merit but does not fully meet PLOS ONE’s publication criteria as it currently stands. Therefore, we invite you to submit a revised version of the manuscript that addresses the points raised during the review process.

We look forward to receiving your revised manuscript.

Kind regards,

Tinggui Chen

Academic Editor

PLOS ONE

Additional Editor Comments:

I have completed my evaluation of your manuscript. The reviewers recommend reconsideration of your manuscript following major revision. I invite you to resubmit your manuscript after addressing the comments below.

Reviewers' comments:

Reviewer's Responses to Questions

**Comments to the Author**

1. If the authors have adequately addressed your comments raised in a previous round of review and you feel that this manuscript is now acceptable for publication, you may indicate that here to bypass the “Comments to the Author” section, enter your conflict of interest statement in the “Confidential to Editor” section, and submit your "Accept" recommendation.

Reviewer #1: All comments have been addressed

Reviewer #2: All comments have been addressed

2. Is the manuscript technically sound, and do the data support the conclusions?

Reviewer #1: Yes

Reviewer #2: Yes

3. Has the statistical analysis been performed appropriately and rigorously? 

Reviewer #1: Yes

Reviewer #2: Yes

4. Have the authors made all data underlying the findings in their manuscript fully available?

Reviewer #1: Yes

Reviewer #2: Yes

5. Is the manuscript presented in an intelligible fashion and written in standard English?

Reviewer #1: Yes

Reviewer #2: Yes

6. Review Comments to the Author

Reviewer #1: English language results improved.

Several issua have revised and currently have improved specifications.

No more concerns

Reviewer #2: 1. In terms of theoretical framework construction, the paper is slightly thin in theoretical elaboration and model construction. It is suggested that the authors strengthen the sorting and elaboration of related theories to further clarify the connotation and influencing factors of green technology innovation efficiency, so as to better support the subsequent empirical analysis and conclusions.

2. The source and selection of data is critical to the accuracy and reliability of the study results. Authors need to clearly state the reliability of the data sources and how the data were processed to ensure the rigor of the study.

3. In terms of research methodology, the thesis adopts a fuzzy set qualitative comparative analysis method (fs-QCA) based on group theory to explore the complex causal mechanisms and grouping paths that affect the efficiency of green technological innovation among enterprises, governments and markets. The choice of this method is innovative but may have some limitations in practical application. The authors need to analyze the applicability of the method in depth and compare it with other possible methods to verify the robustness of the findings.

4. It is recommended to increase the limitations of the paper.

7. PLOS authors have the option to publish the peer review history of their article (what does this mean?). If published, this will include your full peer review and any attached files.

Reviewer #1: No

Reviewer #2: No

---

## [Author Response · Author response to Decision Letter 2]

5 Jun 2024

Dear reviewers

Thank you for reviewing our manuscript. We appreciate your constructive suggestions and have revised it accordingly. We have now clarified the applicability of the manuscript in more detail and have carefully revised the relevant issues you raised. In addition, we have scrutinized the manuscript and made careful revisions to similar problems that have arisen elsewhere. Thank you again for your valuable suggestions. Below are the answers to each of your valuable comments, and you can see the revised manuscript or manuscript for specific changes.

Q1: In terms of theoretical framework construction, the paper is slightly thin in theoretical elaboration and model construction. It is suggested that the authors strengthen the sorting and elaboration of related theories to further clarify the connotation and influencing factors of green technology innovation efficiency, so as to better support the subsequent empirical analysis and conclusions.

Response: Thank you very much for your suggestions. With your suggestions, we have carefully sorted out the relevant theories to make them more logical; we have also used more ink to construct the model and carefully elaborated on the profound influence of enterprises, governments, and markets on the efficiency of green technological innovation. We hope that this modification can better support the empirical content in the later part of the paper.

Q2: The source and selection of data is critical to the accuracy and reliability of the study results. Authors need to clearly state the reliability of the data sources and how the data were processed to ensure the rigor of the study.

Response: Thank you very much for your suggestion, yes, the source of data is very important, our data in this thesis mainly comes from CHINA STATISTICAL YEARBOOK ON science and technology and CHINA STATISTICAL YEARBOOK ON environment , the former is the Chinese The former is a statistical information book reflecting the situation of science and technology activities in China, edited by the Department of Social, Scientific and Cultural Industry Statistics of the National Bureau of Statistics of China and the Department of Strategic Planning of the Ministry of Science and Technology, which includes the annual statistics on science and technology of 31 provinces, autonomous regions, municipalities directly under the central government, and the relevant departments of the State Council of China, while the latter is an annual comprehensive statistical information book reflecting the basic situation of China's environment, edited by the National Bureau of Statistics of China, the Ministry of Ecological and Environmental Protection, and other relevant ministries. The latter is an annual comprehensive statistical information reflecting the basic situation of various fields of the environment in China, edited by the National Bureau of Statistics of China, the Ministry of Ecology and Environment, and other relevant ministries. The source of data is reliable. Due to the large amount of data involved, the processing of the data is placed in Section 5.1.2, Selection of Indicators, and 5.2, Conditional Variable Measurement of the article.

Q3: In terms of research methodology, the thesis adopts a fuzzy set qualitative comparative analysis method (fs-QCA) based on group theory to explore the complex causal mechanisms and grouping paths that affect the efficiency of green technological innovation among enterprises, governments and markets. The choice of this method is innovative but may have some limitations in practical application. The authors need to analyze the applicability of the method in depth and compare it with other possible methods to verify the robustness of the findings.

Response: Thanks to your suggestion, we have innovatively chosen the QCA method to explore the complex causal mechanisms of green technological innovation, and this method has been selected after much deliberation. In section 4.1 of the thesis, we have analyzed the applicability of this method in depth. At your suggestion, we have conducted three stability tests in section 6, Robustness Tests, using the two main methods: the Adjustment Consistency thresholds, adjusting calibration anchors, and grouping analyses across time periods to make our findings more stable.

Q4: It is recommended to increase the limitations of the paper.

Response: Thank you very much for your suggestion. We recognized the article's limitations at your suggestion, and in section 7, we have added a description.

Dear reviewer, thank you again for your valuable suggestions, which made our article more full and readable, and the quality of the article has been taken to the next level. Thank you very much, and have a nice day!

Thank you very much for your attention and time.

Your sincerely

Li Zhang

Shenyang University of Technology

---

## [Decision Letter · Decision Letter 3]

17 Jun 2024

PONE-D-23-35015R3Research on the Enhancement Path of Green Technology Innovation Efficiency under the Group PerspectivePLOS ONE

Dear Dr. Zhang,

Thank you for submitting your manuscript to PLOS ONE. After careful consideration, we feel that it has merit but does not fully meet PLOS ONE’s publication criteria as it currently stands. Therefore, we invite you to submit a revised version of the manuscript that addresses the points raised during the review process.

We look forward to receiving your revised manuscript.

Kind regards,

Tinggui Chen

Academic Editor

PLOS ONE

Journal Requirements:

Reviewers' comments:

Reviewer's Responses to Questions

**Comments to the Author**

1. If the authors have adequately addressed your comments raised in a previous round of review and you feel that this manuscript is now acceptable for publication, you may indicate that here to bypass the “Comments to the Author” section, enter your conflict of interest statement in the “Confidential to Editor” section, and submit your "Accept" recommendation.

Reviewer #1: All comments have been addressed

Reviewer #2: All comments have been addressed

2. Is the manuscript technically sound, and do the data support the conclusions?

Reviewer #1: Yes

Reviewer #2: Yes

3. Has the statistical analysis been performed appropriately and rigorously? 

Reviewer #1: Yes

Reviewer #2: Yes

4. Have the authors made all data underlying the findings in their manuscript fully available?

Reviewer #1: Yes

Reviewer #2: Yes

5. Is the manuscript presented in an intelligible fashion and written in standard English?

Reviewer #1: Yes

Reviewer #2: Yes

6. Review Comments to the Author

Reviewer #1: The manuscript is greatly improved in its presentation,

several paragraphs are now better explained and more understandable

No more concerns

Reviewer #2: I find that authors have put considerable effort into addressing my comments. As a result, the paper is very much improved and I have no problem in recommending it for publication. Meanwhile, I recommend that authors pay attention to the language fluency of their papers, avoid overly long or complex sentences, and simplify the presentation to improve readability.

7. PLOS authors have the option to publish the peer review history of their article (what does this mean?). If published, this will include your full peer review and any attached files.

Reviewer #1: No

Reviewer #2: No

---

## [Author Response · Author response to Decision Letter 3]

21 Jun 2024

Dear reviewers

Thank you for reviewing our manuscript. We appreciate your constructive suggestions and have revised the manuscript accordingly. We have now clarified the applicability of the manuscript in more detail and have carefully revised the relevant issues you raised. In addition, we have scrutinised the manuscript and made careful revisions to similar issues that have arisen elsewhere. 

We really appreciate your valuable comments to us, this article has been greatly improved and a big part of it is due to you, thank you again. We have touched up the whole article and revised the long and difficult sentences that are not easy to understand and checked the grammatical statements of the whole article in the hope that it will better meet the reading needs of the readers.

Dear reviewer, thank you again for your valuable suggestions, which made our article more full and readable, and the quality of the article has been taken to the next level. Thank you very much, and have a nice day!

Thank you very much for your attention and time

---

## [Decision Letter · Decision Letter 4]

26 Jun 2024

Research on the Enhancement Path of Green Technology Innovation Efficiency under the Group Perspective

PONE-D-23-35015R4

Dear Dr. Zhang,

We’re pleased to inform you that your manuscript has been judged scientifically suitable for publication and will be formally accepted for publication once it meets all outstanding technical requirements.

Kind regards,

Tinggui Chen

Academic Editor

PLOS ONE

Additional Editor Comments (optional):

Reviewers' comments:

Reviewer's Responses to Questions

**Comments to the Author**

1. If the authors have adequately addressed your comments raised in a previous round of review and you feel that this manuscript is now acceptable for publication, you may indicate that here to bypass the “Comments to the Author” section, enter your conflict of interest statement in the “Confidential to Editor” section, and submit your "Accept" recommendation.

Reviewer #1: All comments have been addressed

Reviewer #2: All comments have been addressed

2. Is the manuscript technically sound, and do the data support the conclusions?

Reviewer #1: Yes

Reviewer #2: Yes

3. Has the statistical analysis been performed appropriately and rigorously? 

Reviewer #1: Yes

Reviewer #2: Yes

4. Have the authors made all data underlying the findings in their manuscript fully available?

Reviewer #1: Yes

Reviewer #2: Yes

5. Is the manuscript presented in an intelligible fashion and written in standard English?

Reviewer #1: Yes

Reviewer #2: Yes

6. Review Comments to the Author

Reviewer #1: The manuscript is greatly improved in its presentation,

several paragraphs are now better explained and more understandable

No more concerns

Reviewer #2: The author has revised the paper in accordance with the comments and the paper has been greatly improved. I have no further comments and recommend the article for publication.

7. PLOS authors have the option to publish the peer review history of their article (what does this mean?). If published, this will include your full peer review and any attached files.

Reviewer #1: No

Reviewer #2: No

---

## [Editor Report · Acceptance letter]

4 Jul 2024

PONE-D-23-35015R4 

PLOS ONE

Dear Dr. Zhang, 

I'm pleased to inform you that your manuscript has been deemed suitable for publication in PLOS ONE. Congratulations! Your manuscript is now being handed over to our production team.

Kind regards, 

on behalf of

Dr. Tinggui Chen 

Academic Editor

PLOS ONE